# Understanding dynamics of private tuberculosis pharmacy market: a qualitative inquiry from a South Indian district

Vijayashree Yellappa  ,[1,2] Himabindu Bindu,[1] Neethi Rao,[1] Devadasan Narayanan[1]

[1]Health Service Delivery, Institute of Public Health Bengaluru, Bangalore, Karnataka, India
[2]PPP Division, NITI Aayog, Delhi, Delhi, India

**Correspondence to**
Dr Vijayashree Yellappa;
vijayashreehy4@gmail.com

## ABSTRACT

**Objectives** In India, retail private pharmacists (RPPs) are often patients' first point of contact for diseases, including tuberculosis (TB). We assessed the factors influencing RPPs' referral of patients with chest symptoms to the National TB Elimination Programme (NTEP) and the way business is carried out with reference to TB drugs.

**Design** We conducted semistructured interviews with a purposive sample of 41 RPPs in a South Indian district between May and October 2013. Data were collected from urban areas (21 RPPs) and rural areas (20 RPPs) employing the principle of data saturation. Data were analysed thematically using NVivo V.9.

**Results** Knowledge and compliance of RPPs regarding TB symptoms and regulatory requirements were found to be poor. The RPPs routinely dispensed medicines over the counter and less than half of the respondents had pharmacy qualifications. None of them had received TB-related training, yet half of them knew about TB symptoms. Practice of self-referrals was common particularly among economically poorer populations who preferred purchasing medicines over the counter based on RPPs' advice. Inability of patients with TB to purchase the full course of TB drugs was conspicuous. Rural RPPs were more likely to refer patients with TB symptoms to the NTEP compared with urban ones who mostly referred such clients to private practitioners (PPs). Reciprocal relationships between the RPPs, PPs, medical representatives and the prevalence of kickbacks influenced RPPs' drug-stocking patterns. PPs wielded power in this nexus, especially in urban areas.

**Conclusion** India hopes to end TB by 2025. Our study findings will help the NTEP to design policy and interventions to engage RPPs in public health initiatives by taking cognisance of symbiotic relationships and power differentials that exist between PPs, RPPs and medical representatives. Concurrently, there should be a strong enforcement mechanism for existing regulatory norms regarding over-the-counter sales and record keeping.

## INTRODUCTION

Tuberculosis (TB) is the leading infectious killer globally. Ten countries accounted for 75% of the cases, where India and China accounted for 39% of the global gap.[1] Though

**Strengths and limitations of this study**

► Work was undertaken to study the phenomenon naturalistically in the implementation setting, thereby providing sound empirical evidence on retail private pharmacists' (RPPs) practices.

► The study has employed a rigorous qualitative research approach that revealed the reciprocal relationships between RPPs, private practitioners and medical representatives influencing RPPs' drug-stocking patterns.

► Some of the initially selected pharmacists refused interviews so the data available from this study may not be fully representative of the community.

► The study draws findings based on self-reporting by RPPs, which may not necessarily correspond to their actual practice.

► We considered only RPPs registered with district drug controller, where many non-registered pharmacies also dispense drugs over the counter.

the government of India offers free quality-assured TB diagnosis and treatment through the National TB Elimination Programme (NTEP), which was earlier known as Revised National TB Control Program,[2] more than half of TB cases are managed by private practitioners (PPs) in the country.[3 4] Evidence shows that PPs rarely follow standard TB management guidelines[5 6] and thus pose a threat of severe forms of drug-resistant TB.

In most parts of Asia, retail private pharmacists (RPPs) are often patients' first point of contact within the healthcare system,[7 8] and they tend to dispense cough syrups, antibiotics and anti-allergic medicines to patients with chronic cough without physician prescription and rarely refer them for TB testing.[9 10] India has about 630 766 RPPs constituting an important part of the private health sector[11]; and for many patients, pharmacies may be their first point of contact, where most drugs, including

antibiotics, can be purchased over the counter.[12] Studies have found that 83% of the surveyed RPPs received up to five prescriptions of anti-TB drugs weekly,[13] a finding reinforced by other studies which assessed the size and characteristics of private sector anti-TB drug sales in India.[14]

Evidence shows that those with chest symptoms who sought care from RPPs at the first instance are more likely to have long diagnostic delays.[15] Nearly half of the RPPs do not refer those with chest symptoms and thus contribute to delays in diagnosis and treatment.[12] Early diagnosis and treatment initiation are crucial to break the chain of transmission of TB in the community. Delays in the diagnosis increase the chances of complications and mortality. It is therefore argued that RPPs could play an important role in the early detection of TB cases by facilitating patient pathways to TB care,[16 17] but it is not the case now.

The NTEP has been committed to providing free, high-quality TB care to patients managed in private health sector through public–private mix (PPM) strategy.[18 19] In 2012, the concept of PPM was expanded to RPPs after successful pilots in Mumbai in collaboration with an Indian pharmaceutical association. Presently, the government is expected to train RPPs to identify and refer those with chest symptoms to the NTEP and provide directly observed treatment short course.[20] RPPs receive no payment for providing such referral services. Engagement of RPPs is important not only to improve TB detection and care, but also limit the abuse of antibiotics. With this background, in 2013, the government introduced the Schedule H1 as an amendment to the Drugs and Cosmetics Rule of 1945, with the intent to control rampant misuse of antibiotics through over-the-counter dispensing.[21] This mandates the chemist to maintain a separate register where the identity of the patient, contact details of the prescribing doctor and the dispensed quantity of the drug are to be recorded and maintained for at least 3 years. Further, TB was made a notifiable disease in 2012, which mandated private health players to notify patients with TB either diagnosed or treated in a private sector.[22]

There have been studies investigating the potential of RPPs to contribute to TB care.[23–25] Attempts at involving RPPs particularly in TB control activities have not always been successful. There is a dearth of literature from India on the potential of RPPs to contribute to TB control activities. To develop appropriate interventions, it is essential to understand factors that influence RPPs' behaviour and how this could be changed to engage RPPs in TB control activities. With this background, we undertook a study to assess the following: (1) RPPs' referral practices linked to NTEP; (2) stocking and dispensing patterns of anti-TB drugs; (3) clients' TB drug-purchasing patterns and (4) explore the provision of kickbacks to RPPs, if any.

This study was conducted as part of a larger research project to evaluate the results-based financing strategies for TB control in India.

## METHODS

### Study setting
The study was carried out in two sites in Karnataka state, India, For the rural setting, Tumkur district (population of 2.8 million) was considered since 80% of the population in the district resides in villages. For the urban setting, Tumkur city (headquarters of Tumkur district, population of 302 143) and KG Halli (population 44 000), 1 of the 198 administrative wards of Bangalore city, were selected.

Study settings consist of both private and public health facilities. TB services under the NTEP are provided free of cost through government facilities. Structure and functioning of the NTEP are elaborated elsewhere.[26] Patients can avail NTEP services either directly accessing public health facilities or through referrals by PPs/RPPs.

Retail private pharmacies are privately owned and they sell drugs for profit, paid out-of-pocket by the clients. These pharmacies range from high-end big outlets staffed by qualified pharmacists to small, roadside stalls staffed by personnel without formal qualifications but used the license of pharmacists who lent their certificates for money. According to regulations, the minimum qualification for registration as a pharmacist is either a diploma or degree in pharmacy from an institution approved by the Pharmacy Council of India.[27] There are typically two types of private pharmacies: 'attached' are the ones which are attached to a private health facility and 'standalone' with no attachment to a health facility. Patients directly buy medicines from these pharmacies over the counter with or without a valid prescription.

### Study participants and sampling
We targeted 40 semistructured interviews purposively with RPPs, 20 each from rural and urban settings, applying the principle of data saturation. In Tumkur district, we randomly selected RPPs from the list maintained with district drug controller. In KG Halli, we considered all 44 pharmacies that were identified through census. Including both study sites, a total of 77 pharmacies were visited. During the visits, 14 pharmacies were closed and 23 RPPs refused to participate in the study. Overall, a total of 41 RPPs participated in the study (20 from rural and 21 from urban setting; 14 from KG Halli and 7 from Tumkur city). All except three pharmacies included in the study were standalone stores.

### Data collection
Data collection happened from May to October 2013. Semistructured interviews were conducted with the staff who dispensed drugs in pharmacies, irrespective of their qualification. The topic guide covered RPPs' referral practices linked with NTEP, stocking and dispensing of TB drugs, clients' TB drug-purchasing patterns and provision of kickbacks. Interview guide was translated to local language, Kannada, and pilot tested before conducting the actual interviews. Information brochure was shared with participants and the objectives of the research were

explained. An appointment was sought and interviews were conducted in the vicinity of pharmacies. Duration of interviews ranged from 30 to 45 min. All interviews were digitally recorded except for four participants who refused audio recording (all from urban setting). Detailed notes were recorded from such interviews.

## Data analysis

Audio-recorded interviews were translated into English and transcribed verbatim by professional transcribers. Data were managed and analysed with the support of QSR NVivo V.9. We conducted a thematic analysis.[28 29] We combined deductive and inductive approaches to analyse the data. The deductive approach was based on the research questions, and new themes emerging from the data were included (inductive approach).[30] Significant statements relating to the factors influencing RPPs' TB management practices were identified as basic codes. VY, HB and NR devised a coding scheme jointly and this coding scheme was tested on a handful of interviews. These initial codes were then refined and organised at a broader conceptual level into themes by grouping them together.[28] Final coding framework is shown in table 1. In the later stages of data analysis, we explored relationships between the themes, across different categories of participants to identify patterns in the data. To increase the internal validity of the analysis, the coding scheme, the memos and the emerging themes were regularly discussed among the authors. Figure 1 represents the different themes that emerged from the data.

## Patient and public involvement

Patients and the public were not involved in the design, implementation, analysis or dissemination of the study.

## RESULTS

### Participants' characteristics

Average age of study participants was 42 years. Only 18 (43%) of the participants had pharmacy qualification and others were graduates in other disciplines. Pharmacies were open for at least 12 hours starting from 9:00 to 21:00. Rural pharmacies had less number of PPs in the catchment area compared with urban ones, which were seemingly crowded. Nearly all RPPs were aware of professional pharmacists' associations, but only 20% of the qualified RPPs were members of these associations. Details of RPPs' characteristics are provided in table 2.

### RPPs' awareness about TB and the NTEP

Almost all RPPs perceived that incidence of TB is decreasing and it is no longer a problem in the community. None of the respondents had received any TB-related training from the NTEP, yet half of them knew about general symptoms and mode of spread of TB. Major sources of information were friends, mass media and billboards. Twelve RPPs (29%) had no idea about TB. Most RPPs knew about the NTEP, but only one RPP from

**Table 1** Coding framework

| Initial coding framework | Final coding framework |
| --- | --- |
| Qualification, age and number of years of business | Study participants' characteristics |
| Number of health facilities present in the catchment area | |
| Client load | |
| Common ailments for which drugs are dispensed on counter | |
| Membership and participation in the pharmacy association | |
| Perception of TB | RPP's awareness about TB and NTEP |
| Awareness about the NTEP | |
| Whether undergone TB training? If, yes, where and when | |
| Awareness about TB notification and other regulatory norms | |
| Factors that influence client's self-referrals to seek care from RPPs | RPP's practices linked to chest symptoms |
| Type of drugs sold over the counter for chest symptoms | |
| RPP's response to self-referred chest symptoms | |
| Factors influencing RPP's referrals of chest symptoms to health providers and the NTEP | |
| Type of drugs prescribed by PPs for chest symptoms | RPP's stocking patterns of TB drugs |
| Prescription patterns of TB drugs by PPs | |
| Factors influencing RPP's stocking of TB drugs | |
| Profile of patients with TB | Clients' drug-purchasing patterns |
| Perception of problems faced by patients with TB | |
| Cost of TB drugs to patients | |
| Factors influencing patient's purchasing of TB drugs | |
| Factors influencing patient's choice of pharmacies | |
| Profile of the health providers who manage patients with TB | Provision of kickbacks to PPs |
| Factors determining the health provider's choice of pharmacies and use of TB drugs | |
| Provision of kickbacks to referring PPs | |

NTEP, National TB Elimination Programme; PPs, private practitioners; RPPs, retail private pharmacists; TB, tuberculosis.

the rural setting reported to have received information directly from the programme. Seven RPPs from the rural setting were aware of TB notification and they considered referrals to government hospitals to be the extent of their

 

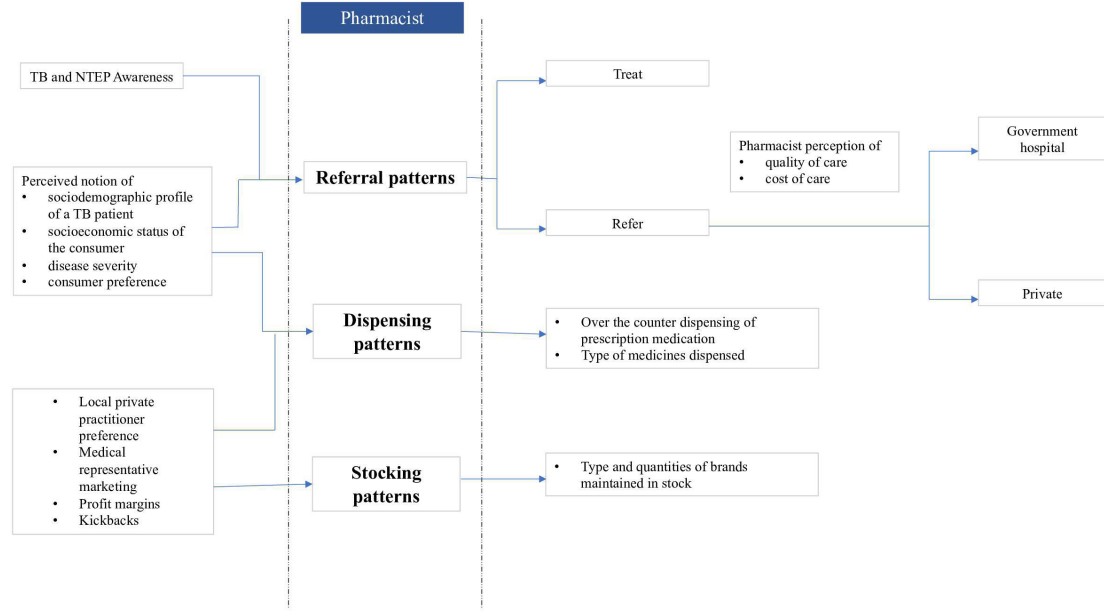

**Figure 1** Different themes that emerged from the data. TB, tuberculosis.

| Table 2 | Demographic data of the RPPs | |
| --- | --- | --- |
| Characteristic | Rural | Urban |
| Gender | | |
| Male | 22 | 16 |
| Female | 1 | 2 |
| Age | | |
| 18–29 | 6 | 3 |
| 30–49 | 15 | 10 |
| 50 and above | 2 | 4 |
| Number of years working at the pharmacy | | |
| 0–5 | 4 | 4 |
| 6–10 | 9 | 9 |
| 11–20 | 6 | 3 |
| >20 | 2 | 0 |
| Not available | 2 | 0 |
| Approximate number of customers per day | | |
| 0–50 | 8 | 8 |
| 51–100 | 8 | 6 |
| >100 | 3 | 2 |
| Not available | 4 | 1 |
| Approximate number of patients with cough per day | | |
| 0 | 1 | 1 |
| 1–15 | 5 | 9 |
| 16–30 | 2 | 1 |
| >30 | 0 | 1 |
| Not available | 15 | 5 |

RPPs, retail private pharmacists.

obligations. None of the RPPs had maintained any kind of records for dispensing TB drugs.

### RPPs' practices in managing patients with chest symptoms

RPPs reported seeing around three to four patients with chest symptoms per day, except six urban RPPs who reported seeing more than 20 patients with chest symptoms per day. Half of the respondents, mostly from the urban setting, reported that they do not dispense drugs without valid prescriptions.

> If they (clients) come here directly, we do not entertain them. If they ask for medicines for small ailments, we give it. Otherwise we inform them to go to the doctor. (U_10)

The remaining half of RPPs, mostly from the rural setting, reported dispensing drugs over the counter for chest symptoms without a prescription. They reasoned that they do this only for patients with cold and cough as it is a common illness, as opposed to diabetes and hypertension which were considered to be serious illnesses.

> We commonly give medicines to such people who are having cold and cough. But for Sugar and BP (hypertension), we have to give specific tablets as prescribed by doctor. Even if they (clients) ask also, we are not supposed to give like that. (R_04)

For self-referred chest symptoms, RPPs dispensed mix of cough syrups, anti-allergic or painkillers, out of which nine RPPs reported dispensing antibiotics to such clients. On further probing, all respondents reported that they will refer patients having cough for more than 15 days to visit nearby doctors for a thorough check-up, except one rural RPP who treated clients himself. Additionally, six respondents (four from rural and two from urban)

reported that they refer such patients for sputum examination and among them two (from rural area) said they would specifically refer such cases to a government facility because of the availability of free quality-assured TB diagnosis there: 'I send patients to the government hospital because the results will be good' (R_18).

> For TB patients the government hospital will provide free medicines. So they do not come outside and purchase medicines. Only patients who have consulted a private doctor, will come for AKT4 (Anti-Koch's Treatment). But, they are very small in number. (R_02)

When probed, majority of the respondents indicated their interest to collaborate with NTEP, if asked in terms of stocking Directly Observed Treatment Short Cours (DOTS) drugs and referring patients for sputum examination.

### RPPs' stocking and dispensing of TB drugs

In total, 78% of respondents reported stocking TB drugs such as AKT3 and AKT4. Stocking of TB drugs was primarily based on the suggestions of PPs practising in the catchment area. One RPP elaborated how PPs influence the stocking of drugs:

> They [doctors] will send prescriptions. Otherwise they will write it as 'keep in stock' and send it. If a TB patient comes to our pharmacy and informs us that he will take medicines here continuously, then we will get that medicine. (R_02)

Another factor influencing stocking of TB drugs was the promotion of certain brands by medical representatives to provide information on profit margins. This heavily influenced RPPs' stocking choices. This phenomenon appeared to be more common and systematic in urban areas where medical representatives tend to persuade PPs and RPPs to stock specific brands of TB drugs.

> It is depending upon the doctors and medical representative's understanding. We look for two to three days. If doctor prescribes the same medicine then we decide to keep that drug. (U_10)

> We decide by calculating profit margins of medicines. Local companies give us more margin than standard companies. Reps give us this information. (U_06)

Rural RPPs hesitated to share information about stocking and dispensing of TB drugs and the number of TB patients purchasing TB drugs compared with urban RPPs. They tend to refer patients to government facilities since TB drugs are available free of cost there.

### Clients' drug-purchasing patterns

Both urban and rural RPPs catered to client load varying from 30 to 300 per day, who commonly purchased drugs for general weakness, diabetes, hypertension and respiratory tract infections. Nearly 60% of RPPs informed that patients tend to directly visit pharmacies without consulting a doctor and others bring old prescriptions to purchase drugs over the counter. Respondents justified the on-counter practice, as this was driven by consumer demand for fast relief of symptoms. They highlighted poverty being the key factor and patients tend to make trade-off to save money and time while seeking care.

> If patients go to a doctor, they have to pay consultation fee of Rs 50 to Rs 100. They are low economic class and cannot afford it. Hence they come here directly. They will get tablets and syrup for the same amount if they come here directly. (U_09)

Patients' choice of pharmacies in the rural setting was based on trust and familiarity with RPPs. However, in the urban setting, proximity to pharmacies, time constraint, unavailability of PPs, lack of availability of medicines in other pharmacies and relatively lower prices influenced patients' choice of pharmacy.

RPPs described patients with TB as those who are in their middle age and financially poor. They estimated average cost of TB drugs per day to be US$7/month and could go as high as US$300/month if nutritional supplements, cough syrup and other antibiotics are combined together in a prescription. Some RPPs deemed the anti-TB drugs to be affordable, while an equal number of them reported TB drugs put a heavy financial strain on patients. More than 85% of respondents asserted that none of the patients with TB purchased the entire course of medication at one time, instead they tend to buy drugs for a few days in one go. On some occasions, patients either tend to reduce the number of drugs prescribed or purchase medicines when they have money. Urban RPPs mentioned that most patients are compelled to take loans for purchasing medicines.

> Only 30%–40% patients will buy 60% of medicines. They are mostly labour class people, doing daily wage work. They get the money only in the evening, hence they buy medicines daily. (U_05)

### Provision of kickbacks to PPs

RPPs having 70–100 clients per day mentioned about the provision of kickbacks to PPs seems to be routine in the urban area, but sporadic in rural. RPPs (21%) narrated a systemic nexus that existed between RPPs, medical representatives and PPs. They estimated that PPs receive commission of about 40% from pharmaceutical companies. Few rural RPPs expressed that the provision of kickbacks to PPs is not a good practice as it has negative impact on their business.

> This is the main problem in the area. From the road till the end, 90% of the doctors are involved in this. Lot of companies are giving them some offers. If the doctor prescribes the particular medicines they get commission up to 40%. (U_06)

RPPs also reported alternate ways through which PPs made profit by owning a pharmacy attached to their clinics

and got a big share in the profits. RPPs were unhappy about this arrangement as this damaged their business.

> Nowadays doctors have their own medical shops. They write such prescriptions which are available with them only. They will not send the patients here, because they (doctors) will be earning commission of 30% to 40%. By chance if they take items from here, they will send it back. (R_08)

Another way of earning kickbacks was to have an understanding with RPPs, and PPs tend to prescribe only such medicines that were available with that particular RPP.

> There are doctors who have adjustments with pharmacists, and they compel patients should go to a specific pharmacy, where they get commissions. (R_10)

When RPPs were asked whether they directly pay kickbacks to PPs, all respondents denied such practice. Some regarded it as unethical with a negative impact on their reputation.

### Limitations of the study

The study draws findings based on self-reporting by the RPPs, which may not necessarily correspond to their actual practice. It is possible that RPPs might be sceptical about reporting the actual number of patients with TB due to fear of scrutiny. Alternatively, this discrepancy could be attributed to socially desirable responses by the participants. To overcome this, we used various approaches including assuring the RPPs of anonymity and confidentiality of the information collected, indirect questioning, probing where RPPs were not particularly forthcoming with the information, among others. However, we acknowledge the potential for eliciting politically correct response and therefore the potential for over/underestimation of the extent of these issues.

Some selected pharmacists refused interviews, so the data available may not be fully representative of the community. We have included only such RPPs who are registered with district drug controller, but there are many non-registered pharmacies that dispense drugs over the counter.

### DISCUSSION

Our study highlights the market dynamics that influence RPPs' referrals patterns about patients with presumptive TB and stocking patterns of anti-TB drugs. Study findings add value to the knowledge on the strategies to involve RPPs in the NTEP. Our study showed only 43% of the respondents had pharmacy qualification and none of them had received any TB-related training. RPPs reported they were not aware about regulatory requirements related to TB, thus the obligatory records and registers required from the government were not maintained. Though RPPs were aware of professional pharmacists' associations, only 20% of the qualified RPPs were members of these associations. RPPs reported 'self-referrals' were common

among patients from economically poorer section, who preferred to purchase drugs over the counter based on an RPP's advice. Majority of RPPs referred clients having cough more than 15 days to nearby PPs. Rural RPPs were more aware of the NTEP and tend to refer TB cases there, far more compared with the urban counterparts. RPPs reported inability of patients with TB to buy full course of TB treatment because of poverty. Our study demonstrates how reciprocal relationships between RPPs, PPs and medical representatives influence RPPs' anti-TB drug-stocking patterns. In general, PPs wielded substantial power in this nexus and received significant kickbacks.

We found that half of the study participants did not have training related to pharmacy, a finding that supports the results from other studies.[31] Hence, it is essential that personnel who dispense drugs should be particularly targeted for public health training irrespective of their qualification. We found RPPs were willing to contribute to TB control activities if asked, a trend reported elsewhere.[32–34] Therefore, a systematic policy of mapping RPPs and orienting them about the NTEP services might prove useful in timely detection of TB cases. Professional associations play a vital role in building the capacity of RPPs. However, our study found only 20% of qualified RPPs had membership with professional organisations, a finding similar to other studies.[35] A policy or an incentive is needed to encourage RPPs to join Indian Pharmaceutical Associations, where they could be systematically trained.

Our study supported the findings of studies which reported the practice of self-referrals for whom drugs were dispensed over the counter,[36 37] which was more prevalent in rural areas. We found rural RPPs were more patronised than urban ones. RPPs who dispensed drugs over the counter justified the practice stating it is mostly driven by consumer demand for fast relief of symptoms as reported in other studies.[38 39] Although some RPPs opined that TB drugs were affordable, patients' purchasing patterns revealed that even seemingly nominal charges could prove to be a heavy financial burden for some patients, confirming other study findings.[40 41] Such voluntary adjustments in drug purchasing by patients to reduce costs may aggravate the risk of drug resistance and lead to poor outcomes. Our study respondents estimated that costs of TB diagnosis and doctor consultations were more expensive than the TB drugs and this could debilitate patients with TB. However, data from the national surveys indicate that the majority of household out-of-pocket expenditures are on drugs, which is at variance from the perceptions of study respondents.[42] This underestimation of the financial burden due to drug costs may influence respondents' behaviour with regard to kickbacks, drug pricing, etc.

Our study demonstrates pharmaceutical nexus operating in the private anti-TB drugs market. Opinions of pharmaceutical company representatives influenced stocking and sale of anti-TB drugs. This resonates the findings from a study that explored the intense

competition within the pharmaceutical industry and the key role played by medical representatives to influence PPs' prescriptions.[43–45] This finding hints at the possibility exploring a business model for subsidising private anti-TB drugs with pharmaceutical industries and also training pharmaceutical company representatives about the NTEP provisions.

PPs not only influenced RPPs' TB drug-stocking practices, but also wielded a lot of power, by forcing RPPs to stock medicines of their choice. This supports the finding of a study from India which showed that PPs receive kickbacks from laboratories and pharmacies (30%).[46] This points towards formulating a regulation that forbids practice of kickbacks. The government of Maharashtra has introduced a bill—Prevention of Cut Practice in Health Care Services Bill, 2017.[47] Evaluation of the effectiveness of this law would be helpful to curb the perverse incentives engendered through the practice of kickbacks.

None of the RPPs notified TB cases nor they maintained any mandatory registers in spite of regulatory requirement. Data sharing on anti-TB drugs from pharmacies in India continues to be poor.[48] There are mixed findings about the effectiveness of Schedule H1 regulation. Some studies have shown that it has minimised on-counter dispensing of first-line anti-TB drugs,[49 50] but another study from South Indian city reported continued irrational dispensing of antibiotics by private pharmacies.[51] Even though suitable laws come into force, their enforcement remains a challenge for an already stretched public health system and drug control authorities.

RPPs are not optimally represented in the national policy discussions in spite of the role played by them as an interface. Though the national strategic plan[18] mentions about the need for engaging chemists, modalities for operationalising this vision is lacking. It is imperative that the symbiotic relations existing between PPs, medical representatives and RPPs should be closely scrutinised for any kind of engagement to meet public health goals. States of Gujarat and Maharashtra have experimented with the setting up of a private provider interface agency to facilitate engagement of RPPs in the NTEP.[52] Investing in public provider support agency[19] that responds to pharmacists' profit making is needed, while promoting the optimal delivery of healthcare services needs to be prioritised.

The NTEP thus needs to adopt multipronged interventions that combine education coupled with regulatory enforcement to engage RPPs in the TB control activities. We have addressed some of these issues in an intervention study conducted in a South Indian district.[53]

**Acknowledgements** Authors would like to acknowledge Mr Ramaiah and Ms Amrutha for facilitating data collection.

**Contributors** VY and DN conceptualised the study. VY collected the data and developed the manuscript. VY, HB and NR coded and analysed the data. HB and NR contributed to the manuscript. All authors have read and approved the manuscript. VY is the first author of the study is responsible for the overall content as guarantor.

**Funding** The work was funded by World Bank (contract no 7165612).

**Competing interests** None declared.

**Patient and public involvement** Patients and/or the public were not involved in the design, or conduct, or reporting, or dissemination plans of this research.

**Patient consent for publication** Not required.

**Ethics approval** This study involves human participants and was approved by IEC-IPH, Bengaluru, India. The institution has a mechanism of a thorough examination of ethical issues of every research project involving human participants. The committee gives elaborate feedback on the ethical issues to be addressed (as seen in the approval letter attached). However, the committee has not provided a reference ID number, but the institution provides the date of the approval as seen in the approval letter. Participants gave informed consent to participate in the study before taking part.

**Provenance and peer review** Not commissioned; externally peer reviewed.

**Data availability statement** Data are available upon reasonable request. Data are available upon reasonable request from the corresponding author.

**ORCID iD**
Vijayashree Yellappa http://orcid.org/0000-0003-0390-7874

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
