## [Reviewer comments · BMJ Open]

ARTICLE DETAILS

TITLE (PROVISIONAL)	Understanding Dynamics of Private Tuberculosis Pharmacy Market: A Qualitative Inquiry from a South Indian district
AUTHORS	Yellappa, Vijayashree; Bindu, Himabindu; Rao, Neethi; Narayanan, Devadasan

VERSION 1 – REVIEW

REVIEWER	Rosalind Miller London School of Hygiene and Tropical Medicine, Global Health and Development
REVIEW RETURNED	10-May-2021

GENERAL COMMENTS	Thank you for inviting me to review this manuscript. It addresses the important, yet often neglected, issue of Indian retail private pharmacists in relation to tuberculosis control, and specifically, their interaction with the Revised National TB Control Programme (RNTCP). I have made some comments for you to consider: Generally, the manuscript needs some editing to ensure the written English is clear. For example, abstract line 51 'how businesses are carried out with special reference to TB drugs'. It is unclear what this means but perhaps could be edited to 'RPP's attitudes and practices with respect to anti-TB drugs'. E.g. one of the results headings 'RPP's practices linked to chest symptomatic' does not read correctly; suggestion: 'RPP practices in relation to managing patients with TB symptoms'. Introduction: - The introduction sets out the problem with RPPs clearly and states that those who seek care at RPPs are more likely to have long diagnostic delays. It would be nice to add a couple of sentences expanding on why late diagnosis is so dangerous and problematic from a public health perspective. - Page 5 line 5 discusses H1 drugs. It is not clear from the description here how these are different to the old schedule H. Perhaps elaborate. First line anti-TB drugs always required a prescription, but this schedule introduced stricter record keeping and stricter penalties for not adhering to these rules? Further, from my understanding the introduction of H1 seems to have had some impact on over-the-counter usage in pharmacies? - It is cited that there is a dearth of literature from developing countries on the potential of RPPs to contribute to healthcare. This is accompanied by a relatively old reference (Smith 2009) yet there have been some efforts made with respect to RPPs and specifically
---

	TB during this time e.g overview articles: Konduri and colleagues conducted a comprehensive review of the 'engagement of the private pharmaceutical sector for TB control' and identified 52 interventions involving retail drug outlets. See Konduri N, Delmotte E, Rutta E. Engagement of the private pharmaceutical sector for TB control: rhetoric or reality? J Pharm Policy Pract 2017;10(1):6. My own work has also reviewed the potential of RPPs and previous efforts to engage them in TB control. See Miller, R and Goodman, C. "Quality of tuberculosis care by pharmacies in low-and middle-income countries: Gaps and opportunities." Journal of clinical tuberculosis and other mycobacterial diseases 18 (2020): 100135. Methods:  - It would be good to clarify the ethics clearance with respect to verbal/written consent. Results:  - The paper reports some interesting findings, especially in relation to lack of RPPs knowledge about the RNTCP (although has this changed since 2013) and the relationship between medical reps and private providers which has not been widely explored in the literature. With respect to practices of RPPs in response to TB symptoms, we know from previous research with RPPs that there is often a large discrepancy between what they say and what they do (and this is reported extensively in the literature for a number of conditions, particularly in response to sale of POMs over the counter). This was mentioned in the limitations, but I feel should be highlighted more. There is a strong chance of social desirability bias. - The provision of kickbacks and nexus that exists with RPPs, PPs and medical reps is interesting, but it isn't clear from the paper exactly how this relates to TB medicines. Would be great to focus specifically on how it affects the sale of these medicines given the focus of the paper. Discussion:  - I noted that the research was conducted in 2013, I think it is important to reflect on whether and how RPPs behaviour/awareness of RNTCP activities may have changed since the interviews were conducted. - It is clear from the work that RPPs are not currently well-informed and an under-utilised resource in the Indian setting. I would love to see you reflect on how pharmacists could best be involved in the RNTCP, based on your discussions with RPPs. What are the challenges that need to be addressed for future initiatives to be successful?
--	--

REVIEWER	Kedar Mehta GMERS Medical College Gotri Vadodara, Community Medicine Department
REVIEW RETURNED	08-Jun-2021

GENERAL COMMENTS	The research question is very interesting and has policy implications. The manuscript is well written. However some major concerns in the manuscript are as below:
--

	1. Title mentions - qualitative study but all findings in abstract are described as quantitative study. 2. Number of participants - 41 selection seems to be biased. 3. RNTCP is renamed as NTEP - so it should be mentioned as NTEP - National TB Elimination Programme formerly known as RNTCP 4. Study has been conducted in 2013 - before 8 years - so the findings may be different in current scenario with many modifications in NTEP programme. 5. COREQ guidelines to be followed for qualitative study - please go through checklist of COREQ guidelines for reporting qualitative studies 6. Recent references are not added in this research area. Thorough literature search is required. References are mentioned of year 2011-2016 - no reference from last 5 year duration. 7. Tables - for sociodemographic profile of participants could be added. 8. Figures - some diagram representing themes / subthemes emerging from thematic analysis - could be added.
--	--

VERSION 1 – AUTHOR RESPONSE

Reviewer -1

Reviewer's comments	Response
General	
Manuscript needs some editing to ensure the written English is clear. For example, abstract line 51 'how businesses are carried out with special reference to TB drugs'. It is unclear what this means but perhaps could be edited to 'RPP's attitudes and practices with respect to anti-TB drugs'. E.g. one of the results headings 'RPP's practices linked to chest symptomatic' does not read correctly; suggestion: 'RPP practices in relation to managing patients with TB symptoms'.	The abstract is rewritten to improve the English. In the results section (page 6 -line 13) the heading 'RPP's practices linked to chest symptomatic' has been replaced with 'RPP's practices in managing patients with chest symptoms'
Introduction	
The introduction sets out the problem with RPPs clearly and states that those who seek care at RPPs are more likely to have long diagnostic delays. It would be nice to add a couple of sentences expanding on why late diagnosis is so dangerous and problematic from a public health perspective.	In the lines page 2/ lines 24-26, we have added "Early diagnosis and treatment initiation are crucial to break the chain of transmission of TB in the community. Delays in the diagnosis increase the chances of complications and mortality"

Page 5 line 5 discusses H1 drugs. It is not clear from the description here how these are different to the old schedule H. Perhaps elaborate. First line anti-TB drugs always required a prescription, but this schedule introduced stricter record keeping and stricter penalties for not adhering to these rules? Further, from my understanding the introduction of H1 seems to have had some impact on over-the-counter usage in pharmacies?	We have added a para on schedule H1 in introduction section in page 3/Line number 6- 10 as “With this background, in 2013 the government introduced the Schedule H1 as an amendment to the Drugs and Cosmetics Rule of 1945, with the intent to control rampant misuse of antibiotics through over the counter dispensing²¹. This mandates the chemist to maintain a separate register where identity of the patient, contact details of the prescribing doctor and the dispensed quantity of the drug are to be recorded and maintained for at least 3 years”. In the discussion section we have added a para to discuss the effectiveness of introduction of H1 in page 10/line number 23-26 as “Data sharing on anti-TB drugs from pharmacies in India continues to be poor⁴⁸. There are mixed findings about the effectiveness of Schedule H1 regulation. Some studies have shown that it has minimized on counter dispensing of first-line anti-TB drugs^{49,50}, but another study from south Indian city reported continued irrational dispensing of antibiotics by private pharmacies⁵¹”.
It is cited that there is a dearth of literature from developing countries on the potential of RPPs to contribute to healthcare. This is accompanied by a relatively old reference (Smith 2009) yet there have been some efforts made with respect to RPPs and specifically TB during this time.	We have added two studies suggested by you. Thank you. We have added two sentences in page 3 from line 13-16 and an article from Indian setting which has explored the potential of RPPs as “There have been studies investigating the potential of RPPs to contribute to TB care^{23,24,25}. Attempts at involving RPPs particularly in TB control activities have not always been successful. There is dearth of literature from India on the potential of RPPs to contribute to TB control activities”.
METHODS	
It would be good to clarify the ethics clearance with respect to verbal/written consent.	We have added two sentences in ethics section: page 11/lines 30-32 “After explaining the confidentiality in local language, written consent to participate in the study was obtained from 26 participants and the remaining opted for verbal consent. Authorization for audio recording the interviews was also sought”.

	In page 11, lines 34-35, we have added a line “the audio files were anonymised. The NVivo database was password protected and was only accessible to the research team”.
Results	
The paper reports some interesting findings, especially in relation to lack of RPPs knowledge about the RNTCP (although has this changed since 2013) and the relationship between medical reps and private providers which has not been widely explored in the literature. With respect to practices of RPPs in response to TB symptoms, we know from previous research with RPPs that there is often a large discrepancy between what they say and what they do (and this is reported extensively in the literature for a number of conditions, particularly in response to sale of POMs over the counter). This was mentioned in the limitations, but I feel should be highlighted more. There is a strong chance of social desirability bias.	In page 11, from lines 1-7, we have added two para in the limitation section to explain the issue of social desirability bias and how it would have influenced the findings of the study as “Alternatively, this discrepancy could be attributed to socially desirable responses by the participants. To overcome this, we used various approaches including assuring the RPPs of anonymity and confidentiality of the information collected, indirect questioning, probing where RPPs were not particularly forthcoming with the information, amongst others. However, we acknowledge the potential for eliciting politically correct response exists and therefore the potential for over/underestimation of the extent of these issues”.
The provision of kickbacks and nexus that exists with RPPs, PPs and medical reps is interesting, but it isn’t clear from the paper exactly how this relates to TB medicines. Would be great to focus specifically on how it affects the sale of these medicines given the focus of the paper	Prevalence of nexus between RPPs, PPs and medical representatives was a study finding, which we have elaborated in the results section on how it influenced the stocking and sale of anti TB drugs under the theme “RPP’s stocking and dispensing of TB drugs” –in page 7 - from line 23. In the discussion section, in the page 10/line 8 we have added the following lines “Opinions of pharma company representatives influenced stocking and sale of anti-TB drugs”. In page 10/ lines from 11-13, we have added “This finding hints at the possibility exploring a business model for subsidising private anti-TB drugs with pharma industries and also training pharma company representatives about the NTEP provisions”. In the page 10/lines from 14-20, we have refined the paragraph as “PPs not only influenced RPP’s TB drugs stocking practices, but also wielded lot of power, by forcing RPPs to stock medicines of their choice. This supports the finding of a study from India which showed that PPs receive

	kickbacks from laboratories and pharmacies (30%) ⁴⁶ . This points towards formulating a regulation that forbids practise of kickbacks. Government of Maharashtra has introduced a bill - Prevention of Cut Practice in Health Care Services Bill, 2017 ⁴⁷ . Evaluation of the effectiveness of this law would be helpful to curb the perverse incentives engendered through the practice of kickbacks”.
Discussion	
I noted that the research was conducted in 2013, I think it is important to reflect on whether and how RPPs behaviour/awareness of RNTCP activities may have changed since the interviews were conducted.	In page 10, lines, 23-26, we have captured the changes as: “There are mixed findings about the effectiveness of Schedule H1 regulation. Some studies have shown that it has minimized on counter dispensing of first-line anti-TB drugs ^{49,50} , but another study from south Indian city reported continued irrational dispensing of antibiotics by private pharmacies”
It is clear from the work that RPPs are not currently well-informed and an under-utilised resource in the Indian setting. I would love to see you reflect on how pharmacists could best be involved in the RNTCP, based on your discussions with RPPs. What are the challenges that need to be addressed for future initiatives to be successful?	We have added the following sentences as below as a reflection from our study findings: In page 9/ line 35-36, we have added a sentence “A policy or an incentive to encourage RPPs to join Indian Pharmaceutical Associations, wherein they could be systematically trained” In page 10/lines from 11-13 we have added a sentence: “This finding hints at the possibility exploring a business model for subsidising private anti-TB drugs with pharma industries and also training pharma company representatives about the NTEP provisions” In page 10/line 29-37, we have added and modified the para as “RPPs are not optimally represented in the national policy discussions in spite of the role played by them as an interface. Though NTEP’s National strategic plan¹⁸ though mentions about the need for engaging chemists, modalities for operationalising this vision is lacking. It is imperative that the symbiotic relations existing between PPs, medical representatives and RPPs should be closely scrutinized for any kind of engagement to meet public health goals. States of Gujarat and Maharashtra have experimented with the setting up of a private provider interface agency to facilitate engagement of RPPs in the NTEP⁵². Investing in public provider support agency¹⁹ that respond to pharmacists’ profit making needs

	while promoting the optimal delivery of health care services need to be prioritised”.
--	---

REVIWER 2

1. Title mentions - qualitative study but all findings in abstract are described as quantitative study.	Abstract has been rewritten as per the suggestion in page 1/lines from 20-43.
2. Number of participants - 41 selection seems to be biased.	We targeted 40 semi-structured interviews, 20 each from rural and urban settings. We applied the principle of data saturation and we continued the data collection until the data was saturated, ie., to the point when no new information was discovered and there was enough information available for data analysis. With this approach we conducted 41 interviews, which represented the contextual factors influencing RPP’s referral practices and anti-TB drugs stocking patterns.
3. RNTCP is renamed as NTEP - so it should be mentioned as NTEP - National TB Elimination Programme formerly known as RNTCP	The term RNTCP is replaced with NTEP, throughout the manuscript.
4. Study has been conducted in 2013 - before 8 years - so the findings may be different in current scenario with many modifications in NTEP programme.	In page 10, lines 29-37, we have added a para to describe this issue: “RPPs are not optimally represented in the national policy discussions in spite of the role played by them as an interface. Though NTEP’s National strategic plan ¹⁸ though mentions about the need for engaging chemists, modalities for operationalising this vision is lacking. It is imperative that the symbiotic relations existing between PPs, medical representatives and RPPs should be closely scrutinized for any kind of engagement to meet public health goals. States of Gujarat and Maharashtra have experimented with the setting up of a private provider interface agency to facilitate engagement of RPPs in the NTEP ⁵² . Investing in public provider support agency ¹⁹ that respond to pharmacists’ profit making needs while promoting the optimal delivery of health care services need to be prioritised”
5. COREQ guidelines to be followed for qualitative study - please go through checklist of COREQ guidelines for reporting qualitative studies	We have revised the reporting as per the SRQR guidelines, thought-out the manuscript, as per the suggestion of editor. We have marked all the items in the checklist, which is attached as supplemental file for your perusal.

6. Recent references are not added in this research area. Thorough literature search is required. References are mentioned of year 2011-2016 - no reference from last 5 year duration.	Recent articles are replaced with older ones in the introduction and discussion section. Reference number added are as follows: 6,12,18,23,24,25,48-51
7. Tables - for sociodemographic profile of participants could be added.	Table containing sociodemographic profile is added with a title – Table 2 Demographic data of the RPPs in page 6.
8. Figures - some diagram representing themes / subthemes emerging from thematic analysis - could be added.	Diagram representing themes / subthemes emerging from thematic analysis is added in page 4.

VERSION 2 – REVIEW

REVIEWER	Rosalind Miller London School of Hygiene and Tropical Medicine, Global Health and Development
REVIEW RETURNED	14-Oct-2021

GENERAL COMMENTS	I enjoyed reading the updated manuscript and believe this paper adds an important contribution. Just a couple of minor things before publication:  - Throughout the manuscript it refers to referrals to the NTEP. The way it is written as 'referral to NTEP' sounds a bit odd so would be great to add the extra 'the' in i.e referral of patients to the NTEP rather than referral to NTEP. -I think the paper could do with one final proof read. There are a few small errors/English corrections that need to be made e.g introduction 3rd paragraph 'Evidence show that chest symptomatic who sought care from RPPs...' should read 'Evidence shows...' -I wonder if you could move your limitations section so that the paper doesn't end on a negative note.
--

VERSION 2 – AUTHOR RESPONSE

Throughout the manuscript it refers to referrals to the NTEP. The way it is written as 'referral to NTEP' sounds a bit odd so would be great to add the extra 'the' in i.e referral of patients to the NTEP rather than referral to NTEP	Thanks for pointing out this. The word 'the' is added before the NTEP, wherever applicable across the manuscript in line 35,72,104,106, 171,173,176,310,322,364,372, 373, and also in the table 1.
I think the paper could do with one final proof read. There are a few	In line 65, the sentence is

small errors/English corrections that need to be made e.g introduction 3rd paragraph 'Evidence show that chest symptomatic who sought care from RPPs...' should read 'Evidence shows...'	corrected as 'evidence shows' and final proof has been done, as per the suggestion.
I wonder if you could move your limitations section so that the paper doesn't end on a negative note.	Limitation section has been moved above the discussion section from line 284 to 297.